# Soft Robot for Inspection Tasks Inspired on Annelids to Obtain Peristaltic Locomotion

**Diego E. Martinez-Sanchez** [1], **X. Yamile Sandoval-Castro** [2], **Nicolas Cruz-Santos** [1], **Eduardo Castillo-Castaneda** [1], **Maximiano F. Ruiz-Torres** [1] and **Med Amine Laribi** [3,*]

1    Centro de Investigación en Ciencia Aplicada y Tecnología Avanzada Unidad Querétaro, Mecatrónica, Instituto Politécnico Nacional, Querétaro 76090, Mexico; dmartinezs1500@alumno.ipn.mx (D.E.M.-S.); nicolas.cruz@brp.com (N.C.-S.); ecastilloca@ipn.mx (E.C.-C.); mruizt@ipn.mx (M.F.R.-T.)
2    Tecnologico de Monterrey, School of Engineering and Sciences, Monterrey 64849, Mexico; yamile.sandoval@tec.mx
3    Department of GMSC, Pprime Institute CNRS, ENSMA, University of Poitiers, 86000 Poitiers, France
*    Correspondence: med.amine.laribi@univ-poitiers.fr

**Abstract:** Soft robotics is a rapidly advancing field that leverages the mechanical properties of flexible materials for applications necessitating safe interaction and exceptional adaptability within the environment. This paper focuses on developing a pneumatic soft robot bio-inspired in annelids or segmented worms. Segmentation, also called metamerism, increases the efficiency in body movement by allowing the effect of muscle contraction to generate peristaltic locomotion. The robot was built using elastomers by the casting technique. A sequence of locomotion based on two stages, relaxation and contraction, was proposed; the contraction stage is actuated by a vacuum pump. The locomotion performances are compared using different elastomers, such as Ecoflex 00-30, Dragon Skin 20, Mold Star 15 Slow, and Mold Star 30. Experimental tests were carried out inside a plexiglass pipe, 1 inch in diameter; a wide range of frequencies was tested for relaxation and contraction stages to evaluate the effect on the speed of the robot.

**Keywords:** soft crawling robot; bio-inspired robot; material characterization; peristaltic locomotion; pneumatic soft robot





## 1. Introduction

Soft robotics is a rapidly growing subfield of robotics that leverages the mechanical properties of flexible materials to design and fabricate robots capable of safe interaction and environmental adaptability. In recent years, this area of robotics has garnered significant attention across multiple disciplines. It offers innovative solutions to intricate challenges faced by traditional robotics, including the adaptation of actuators and end effectors to complex environments, as well as the manipulation of delicate objects [1–4].

Soft robots consist of actuators that can be activated by various signals, including electrical, chemical, magnetic, thermal, and pneumatic stimuli [5,6]. Among these, the most common approach involves the use of air pressurization within different cavities, enabling simple movements, such as expansion, contraction, twisting, or bending. This versatility makes soft robots suitable for a wide range of applications, including medical engineering, manipulation of fragile products, and industrial inspection. However, it is worth noting that the field of industrial inspection lags behind the other applications due to the unique challenge of operating within confined spaces.

Soft crawling robots distinguish themselves from conventional rigid robots by employing soft materials and actuation mechanisms, which enable them to mimic the movements of creatures such as snakes, worms, and caterpillars. Through the use of compliant materials and flexible architectures, these robots possess the ability to bend and adapt to their

surroundings. This unique characteristic empowers them to navigate uneven terrains, traverse through narrow spaces, and interact safety with their environment.

There have already been outstanding accomplishments in the development of soft crawling biologically inspired robots. Researchers have successfully developed robots with locomotional abilities that resemble their biological counterparts' slithering, undulating, or crawling actions.

The field of science has made significant advances, but there are still many obstacles to overcome before scientific study might advance further. Soft crawling biologically inspired robots' exploration of unoccupied places offers fascinating chances to enhance numerous facets of smart design, control, and integration with cutting-edge sensing technology.

Specifically, the two main problems presented in the mechanical design of these types of robots are [7–10]: (1) numerous air inputs or more than one type of input, making it challenging to control the robot, and (2) low robot performance, causing it to not reach functional displacement speeds for an inspection robot. There are several types of locomotion that can be implemented to achieve displacement; however, peristaltic locomotion is one of the most used in the literature. Some authors have studied this type of locomotion: Xiong (2023) [7] presented multi-material hybrid actuators known as MASH actuators, which combine pneumatics and electrostatic adhesion (EA) to achieve enhanced versatility. By introducing an adaptive deformation-limiting layer inspired by the longitudinal muscles of an earthworm, it enables variation in the actuator's length, stiffness, and position. However, this augmentation of control complexity arises from the utilization of different activation types. On the other hand, Sun [8] discusses the challenges in understanding the kinetics of peristalsis, a common motion pattern in limbless animals, due to the lack of suitable physical models. To address this, Sun proposed a multi-chamber vacuum-actuated soft robot inspired by Drosophila larvae, which mimics their crawling behavior. The robot's soft structure, made of hyperelastic silicone rubber, imitates the larval segmental hydrostatic structure. By controlling the vacuum pressure in each segment based on numerical simulations, the soft robots exhibit peristaltic locomotion. Jiang et al. [11] presented the design, fabrication, and characterization of a soft robot that resembles a worm and is meant to operate inside pipes. The robot utilizes pneumatic actuators and is constructed using origami paper-fabric composites, where the paper acts as the skeleton and the fabric serves as the skin of the robot. To achieve peristaltic locomotion, the robot includes a bellow-like structure for extension and a clamp with a Kresling crease pattern at each end for anchoring. In contrast, Liu [12] presented a modular bionic soft robot that can crawl and overcome obstacles. The robot consists of multiple soft modules, each containing two parallel soft actuators. The experimental results showed that the robot achieved a speed of 7.89 mm/s and could overcome obstacles with a height of 42.8 mm when inflated to a pressure of 70 kPa. These proposed soft robots exhibit a complex geometry to achieve locomotion. Furthermore, for this type of robot, it is essential to conduct a proper selection of the manufacturing material to obtain the desired robot's behavior; the most frequently used manufacturing materials are platinum silicons.

This article introduces a bio-inspired soft robot that exhibits peristaltic locomotion, driven by a vacuum-based actuation mechanism. We propose a simplified geometry using only one material. The mechanical design of the robot draws inspiration from the symmetrical cylindrical segments found in annelids' coeloms [13,14]. Based on Tanaka's peristaltic locomotion mechanical model [15], the robot achieves unidirectional movement through a two-stage sequence consisting of relaxation and contraction, actuated by vacuum signals. On the other hand, soft robotics applications offer a diverse range of materials to choose from. In this study, we conduct a mechanical characterization of Smooth On's Ecoflex 00-30, Dragon Skin 20, Mold Star 15 Slow, and Mold Star 30 elastomers; the aim is to identify the optimal material for fabricating our soft robot. To evaluate their performance, we design a segmented actuator probe and numerically analyze its behavior under contraction motion using the finite element method (FEM) with all the characterized materials. Additionally, the soft robot is manufactured using the casting technique, employing Ecoflex 00-30

and Mold Star 15 Slow materials. We conducted experimental evaluations of the robot's performance by varying the configuration of the masses, inspired by biological annelids anchoring, within an acrylic pipe. The objective of this study is to investigate the impact of the mass configuration on the direction of the robot's displacement. Finally, we present the computation of the robot's displacement velocity using computer vision techniques. Moreover, we report the optimization of the locomotion sequence by varying the period and duty cycle of the vacuum pump.

## 2. Soft Crawling Robot Design

This research focuses on the development of a robot that performs displacement into the tubes with a maximum internal diameter of 1 inch. In nature, various animals are observed to move in confined spaces by using different types of locomotion. Three modes of locomotion found in nature were studied for the development of this robot: snake locomotion [16], inchworm locomotion [17], and peristaltic locomotion [15]. A comparison of these locomotion modes based on their mathematical models was conducted to determine their advantages and disadvantages for implementation in a robot with the necessary physical characteristics to achieve the goal of this research.

Peristaltic locomotion is advantageous for movement in tubes because this strategy uses muscle contractions for locomotion. In contrast, the locomotion of nematodes and snakes is based on sinusoidal waves, which make the body of these animals have a different shape, so they need more space to move.

Peristaltic locomotion is a natural locomotor pattern used by elongated soft-bodied invertebrates; earthworms are segmented annelids composed of identical sections called coeloms [18,19]. This pattern involves alternation in circular and longitudinal segments of muscular contraction, as shown in Figure 1a.

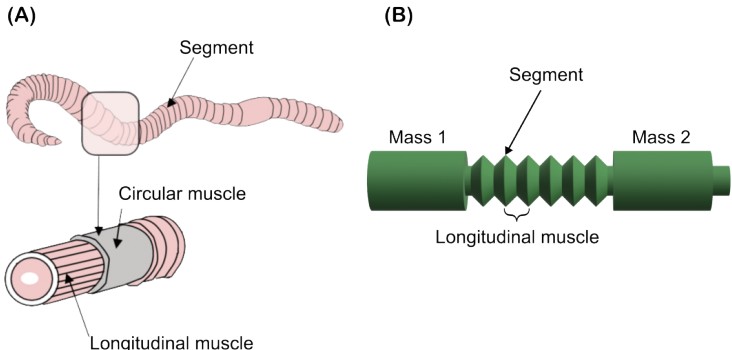

**Figure 1.** (**A**) Schematic of the musculature encompassing the earthworm Crassiclitellata. (**B**) Bio-inspired design of the robot.

Forward movement is produced by the contraction of circular muscles, which extend or elongate the body, while the contraction of longitudinal muscles shortens and anchors the body. Figure 1b shows the design of the bio-inspired robot based on annelids, where the coeloms are represented by the robot segments, which allow the robot's body to contract or extend, such as longitudinal muscles, generating peristaltic locomotion in the soft robot.

The locomotor pattern can be simplified by analyzing two mass blocks $m$ connected through an active element of two springs with constant $k_1$ *and* $k_2$, which are interconnected at the midpoint between the two masses ($m_1$ and $m_2$), allowing for variable opening angle. Figure 2 shows a diagram of the simplification created by Tanaka, and the mechanical model, where the spring is analog to the central body of the robot ($k_{1,2}$).

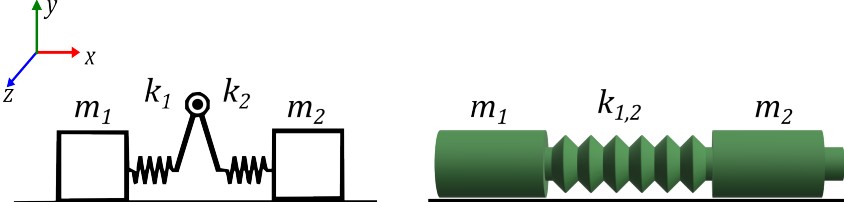

**Figure 2.** Mechanical model of peristaltic locomotion proposed by Tanaka [2].

We propose to study three different configurations in the robots' body, keeping the middle section length but modifying the masses at the ends. Figure 3 shows these models. The first one (A) considers equal mass values $m_1 = m_2$, (B) configuration $m_1 > m_2$, and (C) configuration $m_1 < m_2$ to enhance the displacement towards the larger mass.

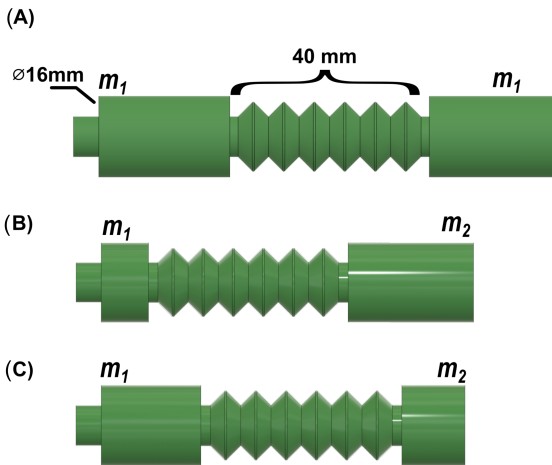

**Figure 3.** The three mass configuration proposals based on Tanaka's mathematical model.

Figure 4 shows the robots' design; it exhibits a sectional cut in the front view, where the dimensions of each segment are described. $R_{ext}$ is the segment's external radius, $\alpha$ is the angle between each segment, $C_d$ is the diameter of the pneumatic chamber, $C_w$ is the crest-to-crest width between segments, and $W_t$ is the thickness of the robot. On the other hand, the minimum coeloms number required to perform peristaltic locomotion is three. However, increasing the number of these segments can enhance the robot's displacement.

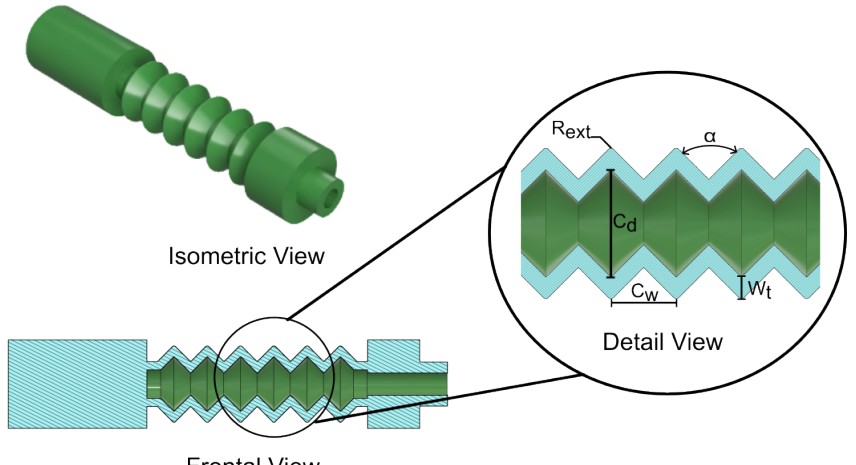

**Figure 4.** Design description of the soft pneumatic robot for peristaltic locomotion.

## 3. Selection of Manufacturing Material

A broad spectrum of materials exists for constructing soft robots, among which Ecoflex 00-30 and Dragon Skin 20 silicons are frequently employed. In this section, we will outline our material selection process based on the maximum contraction achieved in the central body of the robot.

We conducted the characterization of several silicone materials, including Mold Star 15 Slow, Mold Star 30, Ecoflex 00-30, and Dragon Skin 20, all sourced from the manufacturer Smooth On. The characterization process involved uniaxial compression testing following ISO 7743:2017 guidelines [20], uniaxial tension testing in accordance with ISO 37:2011 [21], and pure shear testing. To perform the tests, we fabricated five specimens for each material, following the specifications outlined in the corresponding ISO standards. The TA plus machine (Manufactured by Lloyd Instruments, Bognor Regis, UK) was used for testing, ensuring compliance with ISO 7500-1:2004(E) [22].

Figure 5 illustrates the testing machine, emphasizing its three main sections involved in test development. The initial section is the static segment, comprising a fixed component positioned on the machine's base, where the fixing tools are affixed. On the other hand, the movable section exerts force to deform the specimen. The design of the specimen used in accordance with the aforementioned standards is presented on the right-hand side of Figure 5.

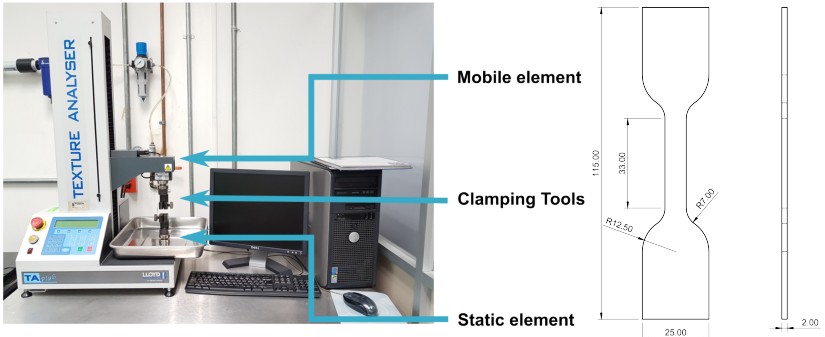

**Figure 5.** TA plus machine used for material testing with highlighted sections and specimen design.

The constitutive models of hyperelastic materials describe their elastic properties under the strain energy function (SEF), which is represented by $W$.

There are different hyperelastic models reported in the literature, which are often used to fit the stress–strain curve of different materials. Most of these models are written in terms of strain invariants, $W = f(I_1, I_2, I_3)$, or in terms of principal stretches, $W = f(\lambda_1, \lambda_2, \lambda_3)$ [23].

Both the Yeoh model and the Mooney–Rivlin model are constitutive models that are used to explain how materials, especially elastomers, behave under deformation. Despite their similarities, the two models differ significantly from one another. The response of the material is described by the Mooney–Rivlin model, an easier isotropic hyperelastic model based on the strain energy density function. It is predicated on the idea that the material exhibits linear behavior in relation to the strain energy density and may be adequately characterized by a polynomial expansion. Two material constants ($C_1$ and $C_2$) are used in the model to describe the behavior of the material, described in Equation (1).

$$W = C_{10}(I_1 - 3) + C_{20}(I_1 - 3)^2 + C_{30}(I_1 - 3)^3 + \sum_{i=1}^{3} \frac{1}{D_i}(J_{el} - 1)^{2i}. \qquad (1)$$

The Yeoh model, on the other hand, is an anisotropic hyperelastic model that provides more flexibility in capturing the behavior of the material. Although higher-order terms are included in the polynomial expansion to account for extra material qualities, the strain energy density function is still used. The Yeoh model adds further material constants

$(C_1, C_2, C_3$, etc.) to precisely depict the material's behavior when subjected to different deformation modes, as described in Equation (2)

$$W = C_{10}(I_1 - 3) + C_{01}(I_2 - 3) + \frac{1}{D_1}(J_{el} - 1)^2. \tag{2}$$

where $I_1$ and $I_2$ correspond to the first and second strain invariants, $C_{10}$ and $C_{01}$ are material-specific parameters, $D_i$ is the material's compressibility constants, and $J_{el}$ is the elastic volume ratio.

The specific material and the degree of accuracy needed for properly describing its deformation response will determine which of the two models is preferable.

The curves obtained by the experimental test are show at the Figure 6, at the left side of the figure we have the curves obtained by the effect of the tension stress on the specimens. At the right side of the same figure the curves show the behavior of the specimens subject to a compression force. Using the values obtained from the experimental tests (tension and compression values) and using Ansys software, the constants of the deviatoric term and the volumetric term were obtained for each of the aforementioned materials. Table 1 shows the coefficients obtained for the Yeoh model as it was the one that best fit the results obtained.

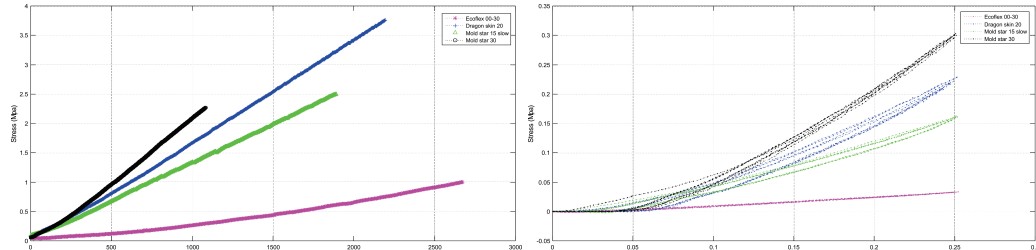

**Figure 6.** Stress–strain graphs of elastomer specimens subjected to tension (**left**) and to contraction (**right**).

**Table 1.** Coefficients of the Yeoh model for each characterized material.

| Material | $C_{10}$ [MPa] | $C_{20}$ [MPa] | $C_{30}$ [MPa] | $D_1$ [MPa$^{-1}$] | $D_2$ [MPa$^{-1}$] | $D_3$ [MPa$^{-1}$] |
|---|---|---|---|---|---|---|
| Ecoflex 00-30 | $1.02 \times 10^{-2}$ | $6.69 \times 10^{-6}$ | $-8.45 \times 10^{-10}$ | 25.43 | 2.38 | $-0.43$ |
| Dragon Skin 20 | $8.09 \times 10^{-2}$ | $7.17 \times 10^{-5}$ | $-9.41 \times 10^{-8}$ | 7.92 | 0.23 | $-0.04$ |
| Mold Star 15 Slow | $5.41 \times 10^{-2}$ | $3.33 \times 10^{-5}$ | $-3.04 \times 10^{-8}$ | 6.57 | 0.41 | $-0.07$ |
| Mold Star 30 | $6.2 \times 10^{-2}$ | $2.48 \times 10^{-4}$ | $-6.74 \times 10^{-7}$ | 6.11 | 0.14 | $-0.02$ |

Once the coefficients were calculated for each material, finite element method (FEM) simulations were conducted in order to analyze the behavior of the robot in the different materials.

Figure 7 depicts the fixed positioning of the robot from the top, specifically for simulations. In the initial state, the red point identified as IP is visible. The first parameter under analysis is the static elongation of the actuator, which signifies the elongation caused by gravity. It is crucial for the actuator to retain its geometry prior to being exposed to an actuating force.

In the relaxed state, the force of gravity is exerted, and the displacement between the two states is measured using the green point RP for all materials. This measurement allows us to determine the static elongation ($\Delta_{S0}$). In the contracted state, a negative pressure of $-10$ KPa is applied, and the contraction relative to the initial state, indicated as $\Delta_{cd}$, is measured using the blue point CP. The results of these simulations are presented in Table 2.

This experiment allows discriminating some materials according to the effect of the gravity during both stages, relaxed and contracted. It can be observed that the Mold Star 15 Slow and Ecoflex 00-30 materials exhibit the highest percentages of contraction (%C)

at the lowest pressure. However, when it comes to static elongation (EE), Ecoflex 00-30 surpasses the other materials due to its superior softness.

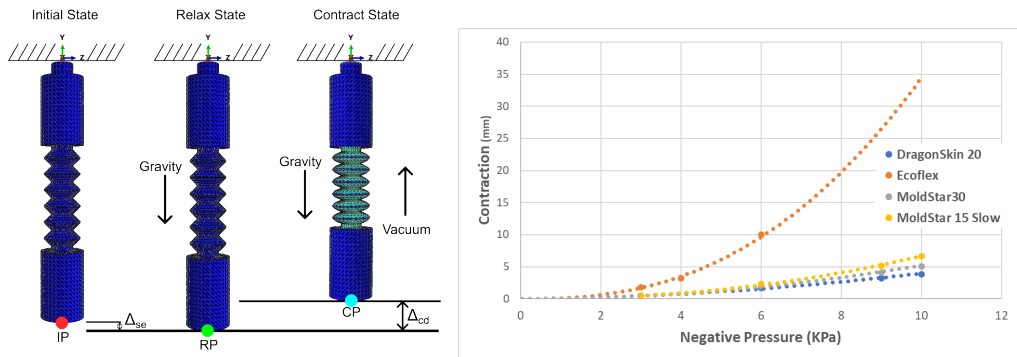

**Figure 7.** Simulations of the robot to evaluate its performance in differents materials.

**Table 2.** FEA modeling results for materials characterized by contraction movement.

| Material | EE [mm] | %C | C [mm] |
|---|---|---|---|
| Ecoflex 00-30 | 4.18 | 22.86 | 8.96 |
| Dragon Skin 20 | 0.67 | 9.84 | 3.86 |
| Model Star 15 Slow | 0.98 | 17 | 6.67 |
| Model Star 30 | 0.83 | 12.97 | 5.09 |

## 4. Manufacturing Process and Locomotion Sequence for the Crawling Soft Robot

The robot was constructed utilizing the preliminary chosen materials in the previous section, Mold Star 15 Slow and Ecoflex 00-30, employing the casting manufacturing technique with molds created through 3D printing. The manufacturing process, which encompasses nine stages, is illustrated in Figure 8:

(1) The main molds are designed to illustrate the equal mass configuration, with molds ($md_2$ and $md_3$) responsible for shaping the robot's external geometry, while molds ($md_1$ and $md_4$) create the internal chamber where pneumatic pressure will be applied. (2) The proposed silicone materials consist of two components: the active component and the catalyst. These two parts are mixed in equal proportions. (3) The material is poured into molds $md_2$ and $md_3$, and any excess material is carefully removed until a uniform layer is attained. (4) The molds are placed inside a vacuum chamber for a period of 30 s to remove any bubbles that may have formed during the mixing process. (5) Molds $md_2$ and $md_3$ are paired with molds $md_1$ and $md_4$, respectively, and securely fastened with screws to ensure uniform pressure distribution during mold insertion. The assembled molds are then left to cure for 6 h for Mold Star 15 Slow and 4 h for Ecoflex 00-30. During this curing process, the halves of the robot are formed. (6) The previous assembly, in this step, is disassembled, and the elastomer is placed around the contour of each cured half. (7) Molds $md_2$ and $md_3$ are assembled to construct the robot and allowed to cure for another 4 or 6 h, depending on the selected manufacturing material. (8) The previous assembly is now disassembled, and the robot is de-molded. (9) Finally, the hose is attached to the air inlet located at one end of the robot.

Drawing inspiration from the principle of peristaltic locomotion, a locomotion sequence was devised for the robot by utilizing its segmented structure, resembling the coeloms of earthworms, as longitudinal muscles. These segments are actuated through the application of negative pneumatic pressure (vacuum), inducing a contraction movement akin to that of an earthworm. The locomotion sequence comprises two alternating steps. The first step, known as the relaxation state, maintains the robot at its initial length without pneumatic feeding. The second step, referred to as the contraction state, involves activating the pneumatic feeding, resulting in the contraction movement in the central bodies' robot. These relaxation and contraction states alternate, enabling the robot to achieve movement

through pneumatic actuation via a single inlet. The direction of movement is determined by the configuration of the masses.

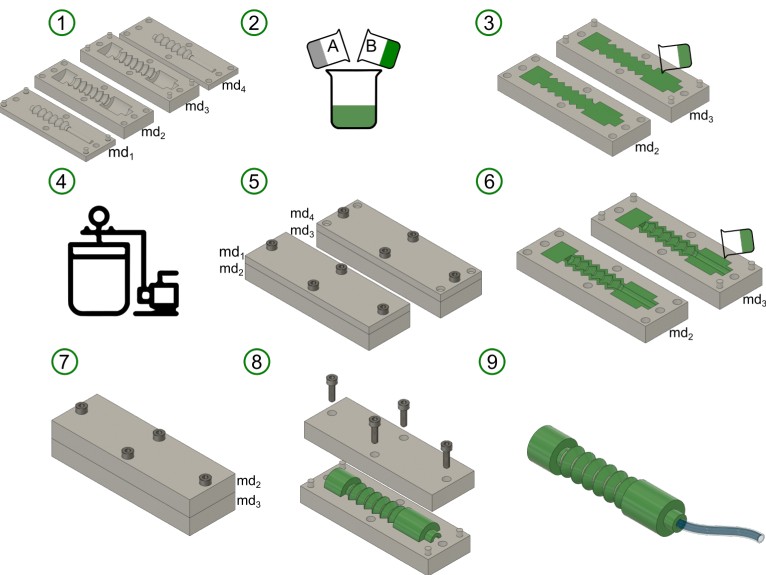

**Figure 8.** Diagram depicting the manufacturing process utilized in the creation of the soft robot.

Figure 9 illustrates the locomotion sequence for the robot's movement on a flat surface, where $\Delta d$ represents the displacement achieved by the robot. The presented locomotion sequence offers the flexibility to easily modify the robot's movement speed by adjusting the relaxation time ($t_1$) and contraction time ($t_2$).

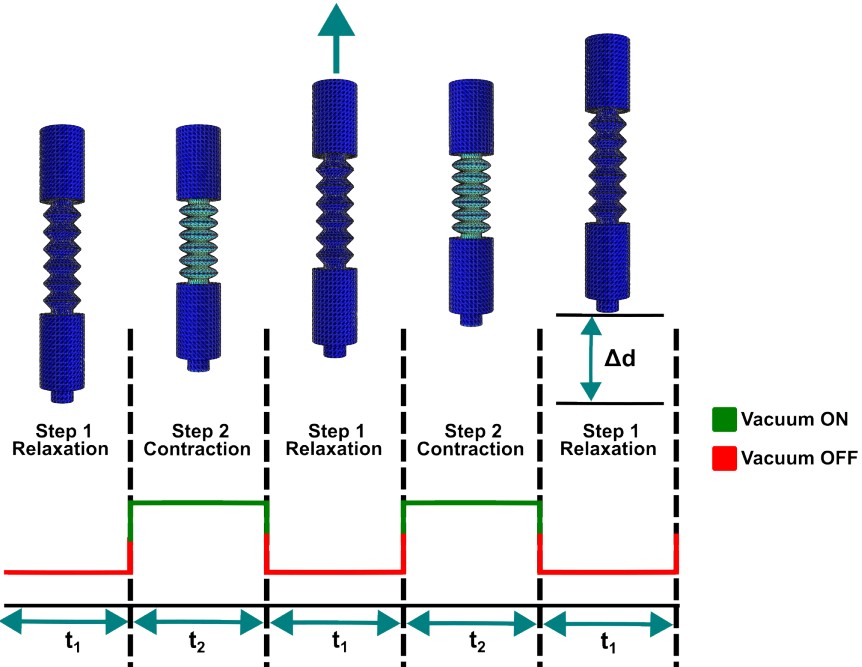

**Figure 9.** Locomotion sequence for robot displacement.

As discussed earlier, the number of coeloms plays a crucial role in the locomotion of the robot. It was mentioned that a minimum of three coeloms is required for the robot's contraction. However, increasing the number of coeloms enables a greater displacement in a shorter time. It is important, though, to avoid excessively increasing the number of coeloms as it can compromise both the contraction capability and the overall geometry

of the soft robot. Figure 10 demonstrates the geometry distortion of the actuator with 10 coeloms caused by gravity.

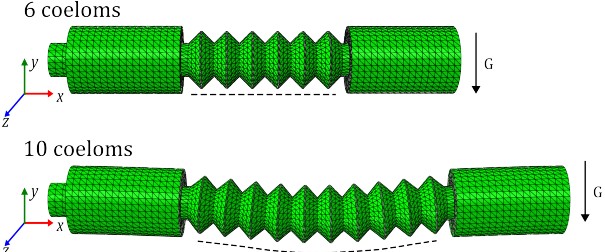

**Figure 10.** Comparison of the effect of the gravity between the robot with 6 and 10 coeloms.

Furthermore, it is necessary to conduct an anchoring analysis to determine the movement direction of locomotion based on the mass configuration of the robot and its corresponding biological model.

The process initiates from an initial or quiescent state, where the worm utilizes its longitudinal muscles to contract until it reaches a maximum point. In contrast, the robot utilizes pneumatic vacuum feeding to generate a contraction force, displacing the masses towards the center of the robot and transitioning it from an initial state to a contracted state.

During this transitional state, the worm employs its anterior appendages to grasp or anchor its body in the direction of its intended displacement. In the case of the robot, the mass with the greater magnitude shifts until it reaches a point where, due to the difference in masses, it can exert a significant force to anchor the robot and cease its motion.

Once the worm reaches maximum contraction, it utilizes its longitudinal muscles to elongate its body, facilitating displacement. Similarly, for the robot, after reaching maximum contraction, the pneumatic feeding is deactivated, enabling the robot to elongate and achieve the desired displacement.

Figure 11 showcases the conducted anchoring analysis. The left-hand side presents a free body diagram, providing information on the normal force of each mass, the contraction force, and its direction in the second stage, along with the elongation force that completes the locomotion cycle. The central and right sections of the figure illustrate the analyzed phases in the locomotion of both the worm and the robot.

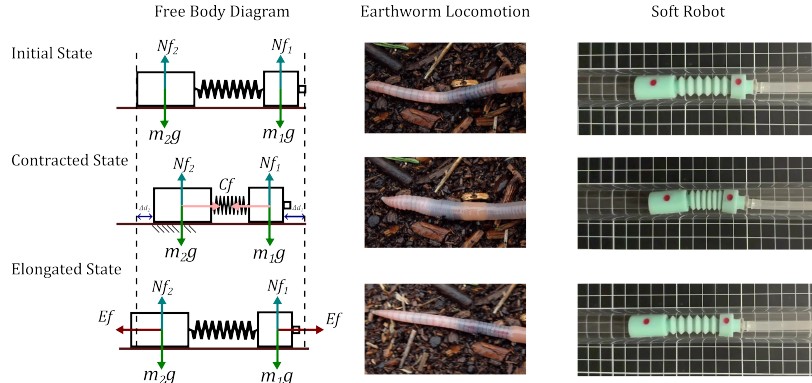

**Figure 11.** Anchoring analysis developed to determine the locomotion direction.

## 5. Experimental Setup

The inspection of confined spaces is a crucial aspect of many industries, where safety and efficiency are paramount in various processes. Confined spaces, including heat exchangers, turbines, and other industrial equipment, pose significant challenges for inspection due to their size, shape, and difficult-to-reach locations.

Inspecting these spaces necessitates specialized expertise and specific equipment to guarantee the safety of workers and the accuracy of inspections. Furthermore, strict

adherence to safety standards and regulations is imperative to mitigate risks to both personnel and the environment.

The inspection of confined spaces within the industry is crucial for maintaining the integrity and efficiency of equipment. In this article, the presented robot specifically addresses the general aspects of confined space inspection. It aims to replicate inspection conditions in a controlled experiment conducted inside a pipe.

In this section, we assess the performance of the locomotion sequence proposed in the previous section. An experiment was conducted to evaluate the displacement of the soft robot within a transparent acrylic pipe with an internal diameter of 20 mm and an external diameter of 24.5 mm. The transparent nature of the pipe enables the study of the robot's behavior inside it and facilitates the determination of displacement speed using computer vision techniques.

The experiment comprises three systems: the control system, computer vision system, and the soft robot. Regarding the control system, shown in Figure 12, it utilizes an Atmel microcontroller (MEGA328P) to generate the required contraction and relaxation times ($t_1$ and $t_2$) for executing the locomotion sequence. The control system includes feedback through the XGZP6847A vacuum sensor, the Adafruit ZR320 vacuum pump capable of providing pneumatic pressure of $-60$ KPa with a volume of 1.8 LPM. Additionally, the system incorporates the MAC 34B-AAA-GDNA-1BA solenoid valve, which facilitates the transition between the negative pressure and ambient pressure supplied to the robot.

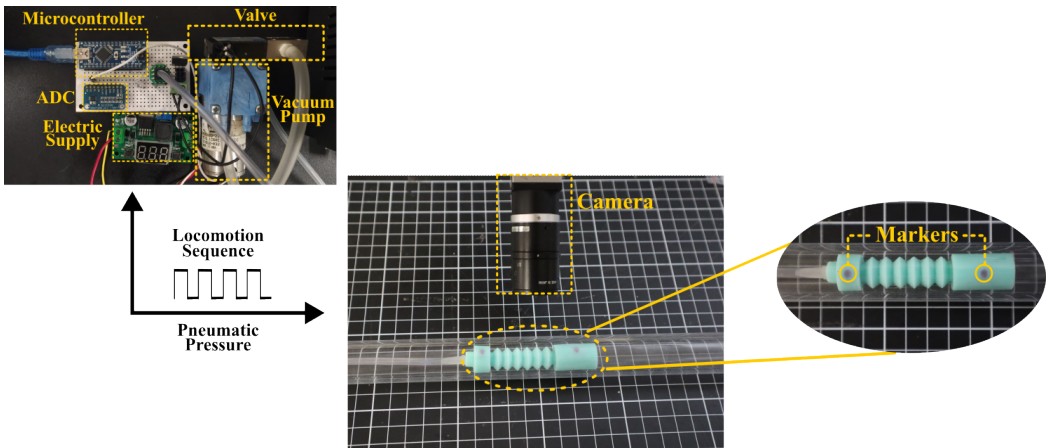

**Figure 12.** Scheme of the experiment with its three constituent systems.

The second system, located in the central part of Figure 12, is the computer vision system. It incorporates an IDS UI-3240CP-C-HQ camera positioned at a distance of 10 cm from the top of the pipe. This camera is responsible for acquiring the necessary images to measure the displacement and speed of the robot.

Finally, the third system depicted on the right-hand side of Figure 12 is the soft robot itself. The robot is presented in detail, and small spheres have been integrated onto the surface of each mass. These spheres serve as markers for image analysis purposes. The robot is positioned within the acrylic pipe, carefully aligned so that the markers on the robot are parallel to the camera plane. Figure 12 provides a comprehensive diagram of the experiment, highlighting each of its individual sections.

## 5.1. Evaluating the Effect of Material on the Locomotion of the Soft Robot

Prior to fabricating the robot, the Mold Star 15 Slow and Ecoflex 00-30 materials were chosen based on the concentration percentage criterion derived from finite element analysis. The aim of this experiment is to assess the performance of these materials inside the pipe, employing the locomotion sequence tracking criterion. Both robots employ a locomotion sequence consisting of a square wave with a period of $T = 2$ s, where $t_1 = t_2 = 1$ s, and a

duty cycle of 50%, over a period of 10 s. The objective is to evaluate which material exhibits superior performance in terms of locomotion inside the pipe.

To analyze the behavior of the robots, a video recording was created for each of them, taking into account the camera calibration method for accurate measurements. The recorded videos were then meticulously analyzed frame by frame to determine the position of the markers placed on the masses of the robots in each image. This analysis provided the planar position and orientation of the robot, enabling the computing of the displacement generated by both robots, denoted as Δd, after each cycle of the locomotion sequence.

Figure 13 illustrates a comparison of the locomotion sequence behavior between the two robots. The green curves represent the square wave pressure generated, where the upper crest corresponds to the contraction stage with a duration of $t_1 = 1$, and the lower crest represents the relaxation stage with a duration of $t_2 = 1$. The red curve, representing the robot fabricated with Mold Star 15 Slow, closely aligns with the proposed reference behavior. The delay observed during the transition between the contraction and relaxation stages is attributed to the time required for the robot to restore its internal pressure to the ambient pressure level.

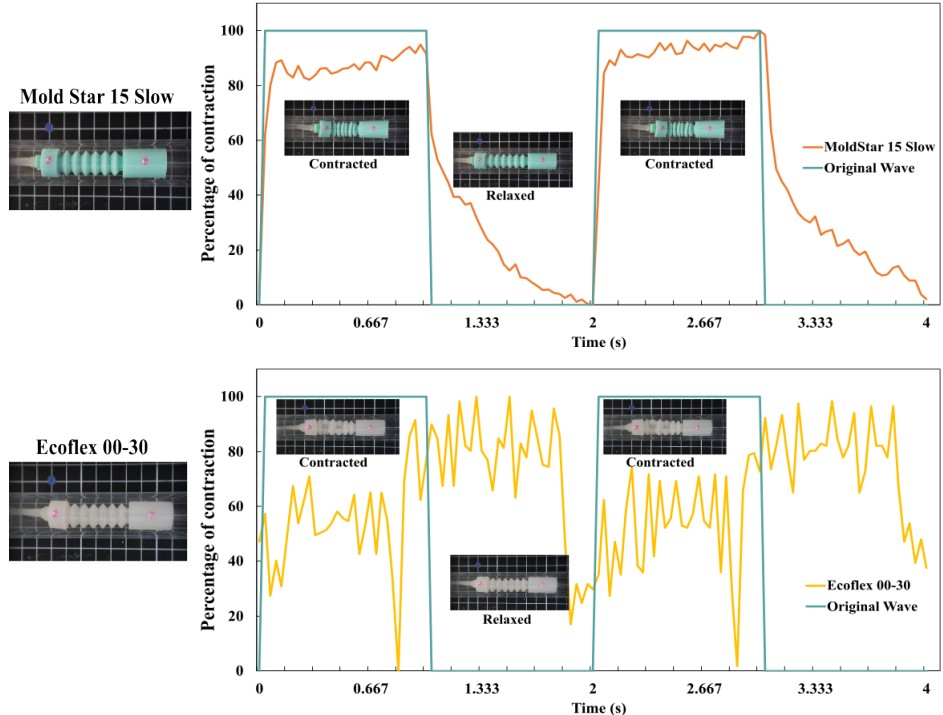

**Figure 13.** Performance of robots manufactured with Ecoflex 00-30 and Mold Star 15 Slow, based on a locomotion sequence utilizing a square wave with a period of $T = 2$ s and a duty cycle of 50%.

The yellow curve, representing the behavior of the robot fabricated with Ecoflex 00-30, exhibits irregular behavior as it deviates from the reference. This discrepancy is attributed to a loss in geometry during the contraction stage. Consequently, it can be concluded that the Mold Star 15 Slow material performs better inside the pipe, demonstrating more consistent behavior.

### 5.2. Effect of Mass Configuration on the Directional Motion of the Robot

The robot fabricated using Mold Star 15 Slow was tested with three different mass configurations, as illustrated in Figure 3. The mass values were set as $m_1 = 5.57$ g and $m_2 = 2.37$ g. It was considered a period of $T = 2$ s and duration of 10 s in the locomotion sequence. Each mass configuration robot was positioned within the acrylic pipe, and the direction of displacement was analyzed using the same technique as the previous experiment.

The behavior observed in each of the mass configuration robots aligns well with the locomotion sequence, where the displacement of each robot is oriented towards the side with the higher mass. In the case of the robot configuration with equal masses, the direction of displacement is influenced by the mass of the tube (15.96 g). Figure 14 visually demonstrates the motion achieved by each mass configuration robot, all directed in a single direction, to showcase their displacement. Considering the future application of this robot in confined space inspection, the selected configuration for further development is the one with $m_1 < m_2$, as illustrated in Figure 14.

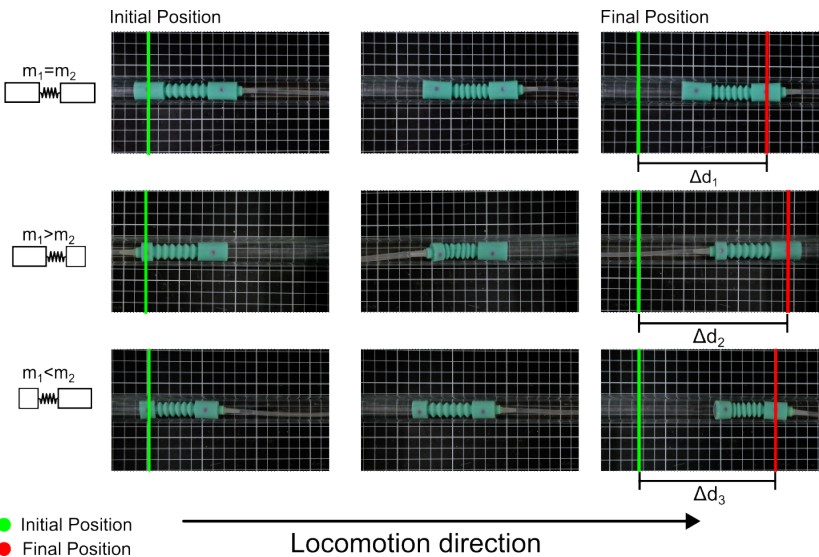

**Figure 14.** Locomotion testing of three mass configuration robots, wherein the direction of locomotion is defined by the greater mass; in the second row, the robot is flipped over the locomotion axis.

The performance of the robot was assessed to determine the optimal period and duty cycle of the locomotion sequence for maximizing its displacement speed. A sweeping analysis of the period of time was conducted, ranging from 0.6 s to 2 s with intervals of 0.2 s. For each period value, four different duty cycle values (25%, 50%, 75%, and 90%) were tested. The robot's performance was evaluated over 10-second intervals, measuring the displacement obtained from each combination by comparing the initial and final positions of the marker placed at $m_1$.

Based on the obtained results, a surface analysis was conducted using the statistical software Design Expert. This analysis enabled the identification of the optimal value that maximized the speed for this specific robot. The results indicated that maintaining a period of 2 s and increasing the duty cycle to 93.10% yield the highest speed.

To validate the maximization results, locomotion tests were conducted by keeping the experiment time at 10 s and analyzing the displacement of the robot. The results of the maximization process were highly encouraging as they demonstrated a significant 48% increase in the robot's speed. The speed improved from 1.33 mm/s, achieved with a duty cycle of 100% and a period of 2 s, to a maximized speed of 5.77 mm/s using the aforementioned optimized values. These outcomes are illustrated in Figure 15.

After analyzing the effects of mass configuration and locomotion sequence variation, a modular robot is proposed. This robot consists of three modules, each actuated by an independent signal. The modular design allows for the creation of a versatile robot capable of traversing in two directions, utilizing the same locomotion principle.

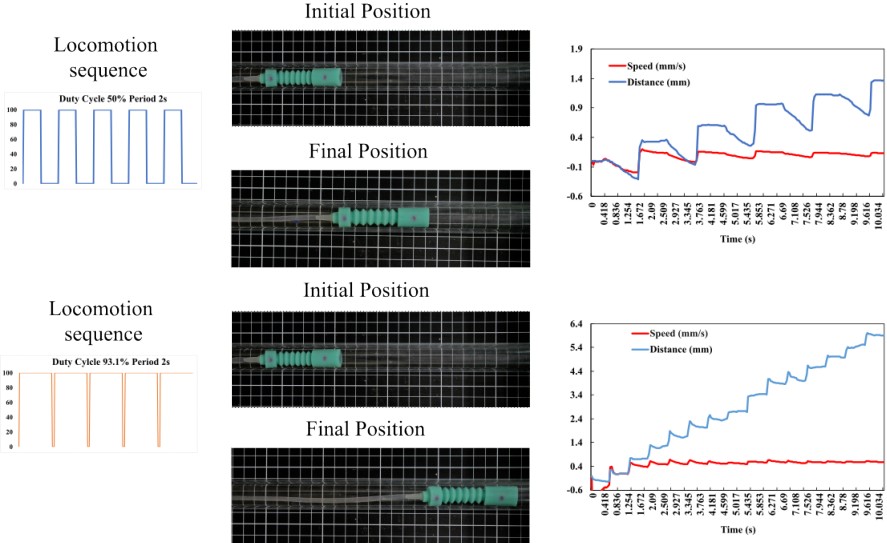

**Figure 15.** Comparison of the velocity and distance generated by the robot using the original locomotion sequence versus the improved locomotion sequence.

Figure 16 exhibits the simulation attained via finite element analysis (FEA) of the proposed design, along with the locomotion sequence requisite to accomplish the robot's movement in both directions.

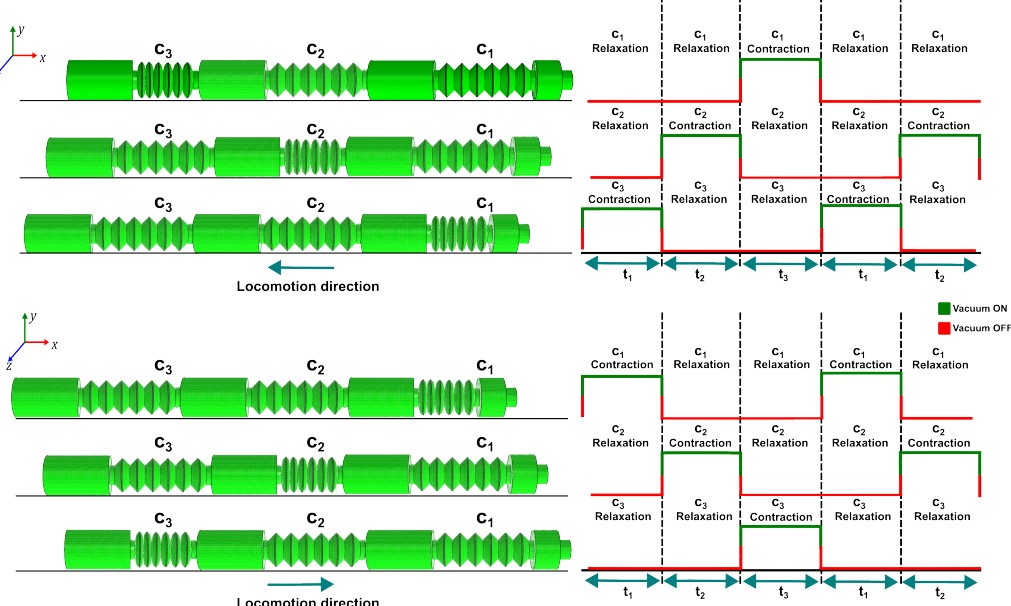

**Figure 16.** Modular robot to obtain locomotion in two directions.

## 6. Conclusions

The inspection of confined spaces is a critical undertaking in a diverse range of industries. These spaces, such as heat exchangers, turbines, or piping, frequently possess limited dimensions and restricted access, thereby rendering the inspection task perilous and arduous.

The segmented design presented in this article is a straightforward bio-inspired approach inspired by earthworms. It utilizes contraction and relaxation movements to achieve unidirectional motion, imitating the actions of longitudinal muscles in worms. This design is identified as the most optimal alternative when compared to other forms of locomotion, such as snake locomotion, inchworm locomotion, and legged locomotion, as it effectively conforms to spatial constraints.

The characterization of platinum silicon materials plays a crucial role in achieving the desired behavior of the robot and provides vital information for conducting simulations through finite element analysis. One particular simulation involved examining the contraction percentage of the robot based on different materials, resulting in the identification of two suitable materials for the robot's construction: Mold Star 15 Slow and Ecoflex 00-30.

The implemented locomotion model in this soft robot, combined with pneumatic actuation, allows for smooth and coordinated movement using a single input control. This feature provides a substantial advantage compared to the works highlighted in the current state of the art. Furthermore, it has been demonstrated that, by following an anchoring analysis, it is possible to selectively change the direction of robot locomotion by adjusting the mass distribution within the robot.

Similarly, the robot's capability to adjust its speed by modifying the duty cycle and period of the pneumatic reference wave feeding offers the flexibility to adapt to various applications.

The future work for this investigation involves conducting an analysis of the robot that takes into account its friction on different surfaces. This analysis will facilitate a comparison between the mathematical model of locomotion and the experimental results. Furthermore, to enhance the capabilities of the robot, we aim to incorporate a second pneumatic input that will enable it to change its direction of movement. This addition will facilitate the implementation of diverse locomotion algorithms and enable the robot to navigate a wider range of environments.

**Author Contributions:** Conceptualization and validation, D.E.M.-S., N.C.-S. and X.Y.S.-C.; material characterization N.C.-S. and M.F.R.-T.; methodology, D.E.M.-S. and M.F.R.-T.; writing—original draft preparation, M.A.L., E.C.-C., X.Y.S.-C and D.E.M.-S.; writing—review and editing, X.Y.S.-C. and D.E.M.-S.; supervision, X.Y.S.-C., M.A.L. and E.C.-C.; project administration, X.Y.S.-C., M.A.L. and E.C.-C. All authors have read and agreed to the published version of the manuscript.

**Funding:** This research received no external funding.

**Data Availability Statement:** The data presented in this study are available in this article.

**Conflicts of Interest:** The authors declare no conflict of interest.

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
