# Peer review of "Soft Robot for Inspection Tasks Inspired on Annelids to Obtain Peristaltic Locomotion"

_machines, doi:10.3390/machines11080779_

Round 1
Reviewer 1 Report
The authors present a pneumatic soft robot bio-inspired by 3 segmented worms allowing the effect of muscle contraction to generate peristaltic locomotion. There are several commnents need to be addressed before any further consideration
1. The authors present two hyperelastic models : the Yeoh model (Eq. 1) and the Mooney-Rivlin model. But there are little further discussion. For example, the pros and cons of each model, and what is the point to introduce these models? can these models be used to describe the locomotion status of the robot?
2. Some figures are poor in quality, especially Fig. 6 and 12, it is hardly to recognize the labels. In Fig. 13, the figures cann't match to each mass configuration.
3. It seems the robot can't move automatically. The movement of the robot is achieved by the relaxation and contraction of the coeloms and the direction is determined by the mass configuration. It means it need an external valve to connect to the robot to provide vacuum and pressure, and the direction of locomotion can not be changed after the mass configuration is fixed, this would be a serious problem for any possible potential application. You do wish the robot to move in any direction with the right stimulus and you do want to avoid any connection to the outerspace when the robot is working in a sealed environment.
Besides the English writing is poor and needs significant improvement.
The English grammar is fine, however extensive work need to be done with the scientific writing.
Author Response
Reviewer 1
The authors present a pneumatic soft robot bio-inspired by 3 segmented worms allowing the effect of muscle contraction to generate peristaltic locomotion. There are several comments need to be addressed before any further consideration
- The authors present two hyperelastic models : the Yeoh model (Eq. 1) and the Mooney-Rivlin model. But there are little further discussion. For example, the pros and cons of each model, and what is the point to introduce these models? can these models be used to describe the locomotion status of the robot?
A further discussion of the two hyperelastic model is introduced in the section 4 (line 155- 165).
On the other hand, the models can be used to estimate the performance of the robot in different motion, such as contraction (for the locomotion sequence).
- Some figures are poor in quality, especially Fig. 6 and 12, it is hardly to recognize the labels. In Fig. 13, the figures cann't match to each mass configuration.
The quality of images 6 and 12 is improved.
Figure 13 was corrected so that the configurations of the mass coincide. In the same way, it was described the meaning in the second row.
- It seems the robot can't move automatically. The movement of the robot is achieved by the relaxation and contraction of the coeloms and the direction is determined by the mass configuration. It means it need an external valve to connect to the robot to provide vacuum and pressure, and the direction of locomotion can not be changed after the mass configuration is fixed, this would be a serious problem for any possible potential application. You do wish the robot to move in any direction with the right stimulus and you do want to avoid any connection to the outerspace when the robot is working in a sealed environment.
The primary focus of this investigation is the development of a soft robot with peristaltic locomotion, inspired by annelids and featuring simplified geometry. However, the future direction of this research encompasses locomotion in both directions. In Section 5, lines 358-363, an alternative approach is presented to achieve bidirectional locomotion by implementing modules within the robot.
Besides the English writing is poor and needs significant improvement.
The English writing was improved
Reviewer 2 Report
The subject of the paper is of great interest in developing original mobile robots for inspection task, exploiting the multiple advantages of bio-inspiration. Please consider following suggestions:
- page 1, lines 31- 33: “Four studies that propose this type of locomotion are Ge (2019) [7], Xavier (2019) [9], Das (2020) [11], and Zhang.” Please include the number of the last reference from your references list; as well as on page 2, line 38; I have not identified this reference in the References list;
- page 3: please correct the Figure 1 caption (now appears to be the text from the journal template);
- please improve the quality of the figures (especially figures 5, 6, 7, 11, 14);
- you didn’t refer to the reference [23] from your list; please refer to it in the text or remove from the References list;
- although the list of references contains 23 literature sources, 19 of these are from 2011-2020, there is no reference from the last 3 years; please try to present similar recent studies in this field and compare your results with more recent results;
- please explain in more detail the principle of locomotion of the studied system; according to figure 9, page 8, why in STEP 2 - Contraction, the posterior segment moves in the direction of movement (why the anterior segment remains fixed), and why in next STEP - Relaxation, the anterior segment moves in the direction of movement and the posterior segment remains fixed? What exactly converts the contraction - expansion of the central segment into unidirectional movement of the system?
- please specify if the connection elements with the vacuum pump influence the behavior and performance of the studied system, especially if the system has to move over long distances ;
-did you consider the influence of the environment (in confined spaces, such as heat exchangers, turbines, and other industrial equipment,) on th material properties?
- minor English corrections are required.
Just minor English corrections are required.
Author Response
Reviewer 2
The subject of the paper is of great interest in developing original mobile robots for inspection task, exploiting the multiple advantages of bio-inspiration. Please consider following suggestions:
- page 1, lines 31- 33: “Four studies that propose this type of locomotion are Ge (2019) [7], Xavier (2019) [9], Das (2020) [11], and Zhang.” Please include the number of the last reference from your references list; as well as on page 2, line 38; I have not identified this reference in the References list;
The missing citation corresponding to Zhang is added, and the citation on page 2 is corrected.
- page 3: please correct the Figure 1 caption (now appears to be the text from the journal template);
The caption of Fig. 1 is corrected.
- please improve the quality of the figures (especially figures 5, 6, 7, 11, 14);
The quality of Figures 5, 6, 7, 11, and 14 were improved.
- you didn’t refer to the reference [23] from your list; please refer to it in the text or remove from the References list;
The list of references is corrected.
- although the list of references contains 23 literature sources, 19 of these are from 2011-2020, there is no reference from the last 3 years; please try to present similar recent studies in this field and compare your results with more recent results;
The bibliographic review was updated
- please explain in more detail the principle of locomotion of the studied system; according to figure 9, page 8, why in STEP 2 - Contraction, the posterior segment moves in the direction of movement (why the anterior segment remains fixed), and why in next STEP - Relaxation, the anterior segment moves in the direction of movement and the posterior segment remains fixed? What exactly converts the contraction - expansion of the central segment into unidirectional movement of the system?
The locomotion sequence, as depicted in Figure 9, is explained in detail in lines 218-231. Furthermore, an anchoring analysis (incorporated to attending your comments) is conducted to elucidate the direction of movement and the underlying mechanism behind the contraction-expansion of the robot's central body, lines 241-260.
- please specify if the connection elements with the vacuum pump influence the behavior and
performance of the studied system, especially if the system has to move over long distances ;
At present, the robot is being examined in a controlled manner, employing a hose length of 2 meters, which is a distance wherein the robot's performance remains unaffected by this value. The influence of mentioned components shall be scrutinized, taking into account their impact on the ongoing analysis of the robot's friction, line 390-395.
-did you consider the influence of the environment (in confined spaces, such as heat exchangers,
turbines, and other industrial equipment,) on th material properties?
In this study, the environment is not considered during the displacement analysis. However, a study of friction on different surfaces with different materials is being carried out, in the conclusions as part of future work this analysis is discussed, , line 358-364.
- minor English corrections are required.
The English writing was improved
Reviewer 3 Report
The topic of the article is highly topical, as soft robotics applications have been developing rapidly in recent times, which can be seen especially in the field of mobile robotics and manipulative robotics. It is a concept that allows to achieve better movement and strength effects. Positioning systems and grippers are a special application, where these structures are also used. But there is a huge problem with sensory systems and systems for controlling the movements of these soft structures. The design processes and technological processes of the production of such robotic systems are also problematic. For a long time, these soft structures had a problem, mainly in technology, but with the development of 3D printing and casting, these problems were partially solved. However, the control problem is mainly due to the elastic properties of the materials used, and the main problem is the non-linearities and hysteresis that these soft systems have, which significantly affects their behavior. Therefore, I consider the solution of this issue to be very important.
Comments:
1. The introduction of the article describes some problems in the development of these systems and also explains the motivation for creating this work. But even so, it would be appropriate to expand the introduction with an overview of other similar works to make it clear where there is empty space for further scientific research.
2. The authors deal with the construction principle of a soft crawling robot, which is intended for movement in pipelines. The authors used biologically inspired locomotion. This principle is not entirely new, and there are several older works that deal with this type of robot locomotion. This area already contains a number of other similar works, and the authors could provide a better overview of the current state in this area.
3. Next, a simple two-mass model of locomotion is presented. However, the mathematical description of this model is not further described. Therefore, please supplement this model description. It shows what problems need to be solved.
4. The authors further deal with the production technology of this device and the selection of suitable material. It is obvious from the next description that it will be a development of a pneumatic chamber actuator. It also includes a description of experiments with materials that the authors consider for the realization of this device. Images 6 are in very poor quality and are unreadable.
5. The benefit is the solution of a suitable type of material and technology for this robot. The results of the experiments are also in Figure 7, but they are also in very poor quality and low resolution.
6. The topic of the article is interesting, but the authors focused more on the technological side of the problem. During the analyses, they did not solve the mathematical model at all and did not at all address the problem of friction with the pipe wall, which significantly affects the movement of the robot.
7. The article mostly deals with engineering problems and lacks a scientific and research part. The novelty and contribution of the article could be inserted into this article precisely by expanding it by solving the above-mentioned problems.
The article requires quite significant revision. Please process comments into a new version of this article.
Author Response
Reviewer 3
The topic of the article is highly typical, as soft robotics applications have been developing rapidly in recent times, which can be seen especially in the field of mobile robotics and manipulative robotics. It is a concept that allows to achieve better movement and strength effects. Positioning systems and grippers are a special application, where these structures are also used. But there is a huge problem with sensory systems and systems for controlling the movements of these soft structures. The design processes and technological processes of the production of such robotic systems are also problematic. For a long time, these soft structures had a problem, mainly in technology, but with the development of 3D printing and casting, these problems were partially solved. However, the control problem is mainly due to the elastic properties of the materials used, and the main problem is the non-linearities and hysteresis that these soft systems have, which significantly affects their behavior. Therefore, I consider the solution of this issue to be very important.
Comments:
- The introduction of the article describes some problems in the development of these systems and also explains the motivation for creating this work. But even so, it would be appropriate to expand the introduction with an overview of other similar works to make it clear where there is empty space for further scientific research.
The introduction was expanded in order to clarify the empty for the future scientific research (line 35-72)
- The authors deal with the construction principle of a soft crawling robot, which is intended for movement in pipelines. The authors used biologically inspired locomotion. This principle is not entirely new, and there are several older works that deal with this type of robot locomotion. This area already contains a number of other similar works, and the authors could provide a better overview of the current state in this area.
This comment was addressed in conjunction with observation 1.
- Next, a simple two-mass model of locomotion is presented. However, the mathematical description of this model is not further described. Therefore, please supplement this model description. It shows what problems need to be solved.
In order to attend this comment, we incorporate an anchoring analyses to determine the movement direction of locomotion based on the mass configuration of the robot and its corresponding biological model, line 241-259
- The authors further deal with the production technology of this device and the selection of suitable material. It is obvious from the next description that it will be a development of a pneumatic chamber actuator. It also includes a description of experiments with materials that the authors consider for the realization of this device. Images 6 are in very poor quality and are unreadable.
The quality of image 6 was improved.
- The benefit is the solution of a suitable type of material and technology for this robot. The results of the experiments are also in Figure 7, but they are also in very poor quality and low resolution.
The quality of image 7 was improved.
- The topic of the article is interesting, but the authors focused more on the technological side of the problem. During the analyses, they did not solve the mathematical model at all and did not at all address the problem of friction with the pipe wall, which significantly affects the movement of the robot.
An anchoring analysis to explain the locomotion sequence is incorporated. On the other hand, a study of friction on different surfaces with different materials is being carried out, in the conclusions as part of future work this analysis is discussed, line 358-364.
- The article mostly deals with engineering problems and lacks a scientific and research part. The novelty and contribution of the article could be inserted into this article precisely by expanding it by solving the above-mentioned problems.
The article has been expanded based on your feedback.
The article requires quite significant revision. Please process comments into a new version of this
article.
The new version of the article is presented.
Reviewer 4 Report
A peristaltic locomotion is presented, with a simplified Tanaka model used to evaluate the most effective configuration of the two masses m1 and m2.
Next, an experimental method is used to define the type of production material to be used. This section is also not very clear, as Figure 6 cannot be read clearly.
The experimental set-up and results are well explained, where the different materials, the effect of mass and the influence of the duty cycle on the locomotion sequence are compared.
In conclusion, the article is acceptable although not so innovative, especially since only the forward movement without change of direction was analysed, where other parameters probably have different influences.
Author Response
Reviewer 4
A peristaltic locomotion is presented, with a simplified Tanaka model used to evaluate the most effective configuration of the two masses m1 and m2.
-Next, an experimental method is used to define the type of production material to be used. This section is also not very clear, as Figure 6 cannot be read clearly.
The section was clarified in the new version of the article, and the quality of Fig. 6 was improved.
-The experimental set-up and results are well explained, where the different materials, the effect of mass and the influence of the duty cycle on the locomotion sequence are compared.
Thank you for your comment.
-In conclusion, the article is acceptable although not so innovative, especially since only the forward movement without change of direction was analysed, where other parameters probably have different influences.
One of the principal contribution of this article is the bio-inspired design, such as the material choosing by means of mechanical characterization and parameters of gravity effect, contraction, and displacement performance. However, a proposal to perform movement in both direction is presented in section 5, line 358-364.
Next, an experimental method is used to define the type of production material to be used. This section is also not very clear, as Figure 6 cannot be read clearly.
The quality of image 6 was improved.
Round 2
Reviewer 3 Report
The article is much better prepared compared to the first version. The authors added part of the mathematical description of the device's function. The introduction of the article has been extended and revised. The experimental part is also supplemented and the presentation of research results is improved.
The article is significantly better, but there are still errors that can be solved in the proof version of the final article.
Comments:
The graphs in Figures 6 and 7 are still unreadable and of very poor quality. Please correct it in the proof version of the article. The legends, the axis description, and the numbers on the axes cannot be read even after magnification.
Some numerical values are missing spaces between the values and the units of the quantity. It also needs to be corrected in the proof version of the article.
On lines 189, 286, the unit "KPa" is incorrectly stated. It should be like "kPa". It's also bad in Figure 7 too.
Units (line 355, 356) should not be written in italic style.